# Mechanism of genome instability mediated by human DNA polymerase mu misincorporation

Miao Guo[1,2], Yina Wang[1,2], Yuyue Tang[1,2], Zijing Chen[1,2], Jinfeng Hou[1,2], Jingli Dai [1,2], Yudong Wang[1,2], Liangyan Wang[1,2], Hong Xu[1,2], Bing Tian[1,2], Yuejin Hua [1,2✉] & Ye Zhao [1,2✉]

Pol μ is capable of performing gap-filling repair synthesis in the nonhomologous end joining (NHEJ) pathway. Together with DNA ligase, misincorporation of dGTP opposite the templating T by Pol μ results in a promutagenic T:G mispair, leading to genomic instability. Here, crystal structures and kinetics of Pol μ substituting dGTP for dATP on gapped DNA substrates containing templating T were determined and compared. Pol μ is highly mutagenic on a 2-nt gapped DNA substrate, with T:dGTP base pairing at the 3′ end of the gap. Two residues (Lys438 and Gln441) interact with T:dGTP and fine tune the active site microenvironments. The in-crystal misincorporation reaction of Pol μ revealed an unexpected second dGTP in the active site, suggesting its potential mutagenic role among human X family polymerases in NHEJ.

[1] Institute of Biophysics, College of Life Sciences, Zhejiang University, Hangzhou, Zhejiang, China. [2] MOE Key Laboratory of Biosystems Homeostasis & Protection, Zhejiang University, Hangzhou, Zhejiang, China. ✉email: yjhua@zju.edu.cn; yezhao@zju.edu.cn

Accurate DNA replication upon DNA damage by DNA polymerases is a key factor determining genome stability to maintain life. The misinsertion of a nucleotide without proofreading causes base substitutions as well as nucleotide additions and deletions, leading to further genome instability and human diseases[1,2]. To date, humans have been shown to express 17 DNA polymerases, which are recruited to the genome depending on the cell cycle and type of DNA substrate. While replicative DNA polymerases containing exonucleolytic proof-reading activity are responsible for the normal DNA replication process, low-fidelity DNA polymerases concentrate on translesion[3], and repair synthesis[4] of damaged DNA bases; for example, Pol η[5,6] and Pol β[7,8] are well documented to be involved in bulky lesion bypass and base excision repair, respectively. On the other hand, DNA double-strand breaks (DSBs) are required for antibody maturation and meiosis and are induced by exogenous agents such as ionizing radiation[9]. Nonhomologous end joining (NHEJ) is the major repair pathway employed by higher eukaryotes to repair DSBs, which requires specialized DNA polymerases to bridge over DSB ends and fill small gaps prior to ligation[10].

Pol μ is one of three human X family polymerases (Pol μ, Pol λ, and TdT) involved in the NHEJ pathway and directly interacts with NHEJ factors, including Ku proteins[11–15]. Compared with replicative DNA polymerase, Pol μ lacks proofreading activity, with an intrinsically high error rate of $10^{-3}$–$10^{-5}$ for dNTP incorporation[16,17]. Despite its reduced fidelity, Pol μ containing an insertion in Loop1 can fill small gapped DNA substrates[18–21], and a recent study revealed that Pol μ can effectively misinsert dGTP opposite 1-nt gapped DNA containing templating T[22]. This misincorporation further facilitates subsequent NHEJ ligation reaction by the DNA ligase IV/XRCC4 complex, which may ultimately lead to genomic instability during NHEJ repair. Despite the partial overlap function in NHEJ repair, the substrate binding and dNTP incorporation properties of Pol μ differ from those of Pol λ[19,23,24]. Pol μ can replicate DSB substrates containing noncomplementary termini, while paired primer termini are preferred by Pol λ[18]. Moreover, in contrast to the sequential gap filling of Pol λ from the first available 3′ unpaired templating base[25], Pol μ appears to fill the gap using exact 5′ unpaired

templating base, which may easily cause microhomology-directed deletions[19].

In addition to the canonical mechanism for primer extension observed by replicative polymerases, Pol μ can bind $Mn^{2+}$-nucleotide pairs prior to binding substrate DNA[23]. Compared with $Mg^{2+}$, $Mn^{2+}$ is strongly preferred by Pol μ for terminal transferase activity as well as NHEJ efficiency[26,27]. Replacing $Mg^{2+}$ with $Mn^{2+}$ accelerates the Pol μ reaction for correct insertion but at the cost of an elevated misincorporation rate[26,28]. Structural studies have revealed that conformational selection of incoming nucleotide binding is absent with Pol μ, which is able to accommodate a damaged templating base and insert ribonucleotide or 8-oxo-dGTP with no distortion of the active site[19,29–31].

To elucidate the molecular mechanism of Pol μ misincorporation, we aimed to determine crystal structures of the polymerase core domain (with Loop2 deletion) of human Pol μ (Pol μ for short) during the process of dGTP misincorporation opposite the templating T on either 1 or 2-nt gapped DNA substrates as well as the in-crystal misincorporation reaction. Complementing these structures, two residues, Lys438 and Gln441, were mutated, and mutant proteins were crystallized to further investigate their roles in dGTP misincorporation. Steady-state kinetic parameters were measured to confirm the structural observations.

## Results

**Kinetic analysis of dGTP misincorporation by Pol μ.** Since the promutagenic mismatch (T:G) causes potential genome instability, we first determined the catalytic efficiency and nucleotide preference of Pol μ polymerase core domain upon misinsertion of dGTP opposite the templating T on 1 and 2-nt gapped DNA substrates in the presence of either $Mg^{2+}$ or $Mn^{2+}$ in the reaction buffer (Fig. 1). In the presence of $Mg^{2+}$, Pol μ was able to misinsert all three kinds of dNTPs opposite the templating T, with distinct preferences for 1-nt gapped DNA (dTTP ≧ dCTP > dGTP) and 2-nt gapped DNA (dTTP ≧ dGTP > dCTP). In addition to the similar preference, $Mn^{2+}$ increases overall product formation. Notably, a weak band beyond the product of dGTP misincorporation was observed with increased protein

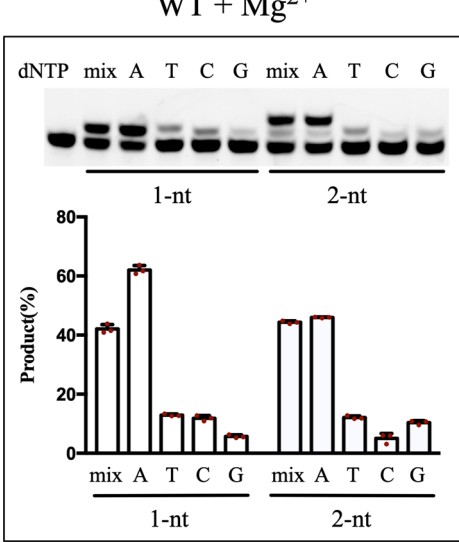

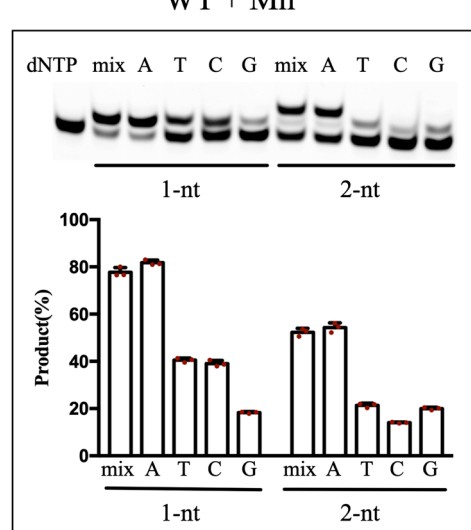

**Fig. 1 Base selection and efficiency of Pol μ incorporation opposite gapped DNA containing one (1 nt) or two (2 nt) templating T in the presence of either $Mg^{2+}$ or $Mn^{2+}$.** Reactions were carried out at a substrate:enzyme molar ratio of 50:1, with a single type or a mixture of all four dNTPs at a total concentration of 0.4 mM at 37 °C for 7 min as described in the "Method" section. A plot of quantification of product formation is shown below the gel (data are presented as mean ± s.e.m, $n = 3$ biologically independent samples).

**Table 1 Kinetic parameters of dGTP misincorporation opposite gapped DNA containing templating T by pol μ.**

| Enzyme | Gap/metal | dNTP | $K_m$ (μM) | $k_{cat}$ (min$^{-1}$) | $k_{cat}/K_m$ (min$^{-1}$ μM$^{-1}$) | $f_{mis}$ * |
|---|---|---|---|---|---|---|
| WT | 1 nt/Mg$^{2+}$ | dATP | 3.1 ± 0.3 | 236.2 × 10$^{-3}$ ± 5.3 × 10$^{-3}$ | 76.6 × 10$^{-3}$ | 6.4 × 10$^{-2}$ |
| | | dGTP | 5.7 ± 0.5 | 28.3 × 10$^{-3}$ ± 0.6 × 10$^{-3}$ | 4.9 × 10$^{-3}$ | |
| | 2 nt/Mg$^{2+}$ | dATP | 17.1 ± 2.7 | 185.8 × 10$^{-3}$ ± 10.6 × 10$^{-3}$ | 10.8 × 10$^{-3}$ | 42.6 × 10$^{-2}$ |
| | | dGTP | 5.0 ± 0.8 | 23.0 × 10$^{-3}$ ± 1.0 × 10$^{-3}$ | 4.6 × 10$^{-3}$ | |
| | 1 nt/Mn$^{2+}$ | dGTP | 5.2 ± 0.6 | 100.1 × 10$^{-3}$ ± 2.6 × 10$^{-3}$ | 19.2 × 10$^{-3}$ | – |
| Q441A | 1 nt/Mg$^{2+}$ | dATP | 9.6 ± 0.8 | 409.1 × 10$^{-3}$ ± 11.3 × 10$^{-3}$ | 42.7 × 10$^{-3}$ | 28.6 × 10$^{-2}$ |
| | | dGTP | 2.0 ± 0.3 | 25.0 × 10$^{-3}$ ± × 0.9 × 10$^{-3}$ | 12.2 × 10$^{-3}$ | |
| K438A | 1 nt/Mg$^{2+}$ | dATP | 9.5 ± 1.2 | 92.2 × 10$^{-3}$ ± 3.5 × 10$^{-3}$ | 9.7 × 10$^{-3}$ | 16.5 × 10$^{-2}$ |
| | | dGTP | 31.1 ± 3.6 | 49.7 × 10$^{-3}$ ± 2.4 × 10$^{-3}$ | 1.6 × 10$^{-3}$ | |
| K438A/Q441A | 1 nt/Mg$^{2+}$ | dATP | 17.6 ± 1.7 | 98.4 × 10$^{-3}$ ± 3.5 × 10$^{-3}$ | 5.6 × 10$^{-3}$ | 80.4 × 10$^{-2}$ |
| | | dGTP | 3.4 ± 0.5 | 15.1 × 10$^{-3}$ ± 0.5 × 10$^{-3}$ | 4.5 × 10$^{-3}$ | |

*$f_{mis}$ is the relative efficiency of misincorporation expressed as the ratio of $[k_{cat}/K_{mdGTP}]/[k_{cat}/K_{mdATP}]$.

**Table 2 Summary of structures.**

| Structure name | Protein | Gapped DNA | Metal | dNTP | PDB Code |
|---|---|---|---|---|---|
| Binary complex | WT | 1-nt gap | Mg$^{2+}$ | – | 7CO6 |
| Complex 1 | WT | 2-nt gap | Mg$^{2+}$ | dGMPNPP | 7CO8 |
| Complex 2 | WT | 1-nt gap | Mg$^{2+}$ | dGMPNPP | 7CO9 |
| Complex 3 | WT | 1-nt gap | Mn$^{2+}$ | dGMPNPP | 7COA |
| Complex 4 | Q441A | 1-nt gap | Mg$^{2+}$ | dGMPNPP | 7COB |
| Complex 5 | K438A/Q441A | 1-nt gap | Mg$^{2+}$ | dGMPNPP | 7COC |
| Post-insertion complex | K438A/Q441A | 1-nt gap | Mg$^{2+}$ | dGTP | 7COD |

concentration (Supplementary Fig. 1), indicating that Pol μ has a weak activity to extend the primer after a T:G mispair in solution.

$K_m$ and $k_{cat}$ were measured for both correct insertion (dATP) and dGTP misincorporation (Table 1). Our results for correct dATP insertions are comparable to published values obtained with DNA containing varied sequence contexts[19]. In the presence of Mg$^{2+}$, the $K_m$ for Pol μ inserting dGTP vs. dATP on a 1-nt gapped DNA substrate increased by ~1.8-fold, and the $k_{cat}$ decreased by ~8.3-fold. When replacing Mg$^{2+}$ with Mn$^{2+}$, the $K_m$ value for dGTP misincorporation was similar, and the $k_{cat}$ increased by ~3.5-fold, indicating that Mn$^{2+}$ enhances the misincorporation reaction of Pol μ in solution. For 2-nt gapped DNA substrate, Pol μ exhibited a much lower catalytic efficiency for the correct base (dATP) insertion in the presence of Mg$^{2+}$ compared with that on a 1-nt gapped DNA substrate (with increased $K_m$ and decreased $k_{cat}$). However, the kinetic parameters for dGTP misincorporation were very similar to those of the 1-nt gapped DNA substrate, resulting in an elevated misincorporation rate of dGTP vs. dATP ($42.6 \times 10^{-2}$) compared with that of the 1-nt gapped DNA ($6.4 \times 10^{-2}$).

**Overview of the Pol μ structures**. The Pol μ core domain (termed WT for brevity) complexed with either 1 or 2-nt gapped DNA and nonhydrolyzable 2′-deoxy-guanosine-5′-[(α,β)-imido]triphosphate (dGMPNPP) opposite the templating T were solved. We determined three types of structures for distinct Pol μ:DNA complexes (Table 2): (1) binary complex: WT crystallized with a 1-nt gapped DNA; (2) ternary complexes: WT with a 2-nt gapped DNA, dGMPNPP, and Mg$^{2+}$ (Complex 1); WT with a 1-nt gapped DNA, dGMPNPP, and Mg$^{2+}$ or Mn$^{2+}$ (Complexes 2 and 3); Q441A single- and Q441A/K438A double-mutant proteins with a 1-nt gapped DNA, dGMPNPP, and Mg$^{2+}$ (Complexes 4 and 5); and (3) post-insertion complex: Q441A/K438A mutant protein with a 1-nt gapped DNA and soaked with dGTP/Mg$^{2+}$. All seven Pol μ complexes crystallized in the $P2_12_12_1$ space group with one

complex per asymmetric unit, and the structures were refined to a resolution between 1.6 and 1.9 Å (Supplementary Table 1). Despite the disordered Loop1 region, the overall structures of these Pol μ:DNA complexes are remarkably similar to previously solved Pol μ ternary complexes[19,29], which adopt a left-hand "closed" conformation (Fig. 2a, Supplementary Fig. 2a). Of the four structural domains of Pol μ, the finger and palm domains hold the upstream primer and template strand, and the thumb and 8-kDa domains interact with the downstream DNA and DNA gap. The incoming nucleotide was observed in all the Pol μ ternary complexes. The pairwise root means square deviation (rmsd) of Pol μ among the seven structures ranged from 0.05 to 0.39 Å over 277–322 Cα atoms, indicating the rigidity of Pol μ during dGTP misincorporation. Moreover, the backbone of downstream DNA and template strand DNA are held tightly by Pol μ and exhibited few differences among all the ternary complexes (Supplementary Fig. 2b), indicating that dGTP misincorporation causes little DNA distortion. Notably, the triphosphate moiety of dGMPNPP and two catalytic metal ions are almost superimposable among all the structures (Supplementary Fig. 2b), which is consistent with the high affinity of dGTP (low micromolar $K_m$ range) during dGTP misincorporation (Table 1).

**dGTP misincorporation on a 2-nt gapped DNA substrate**. The overall structure of WT complexed with 2-nt gapped DNA and dGMPNPP (Complex 1) could be virtually superimposed onto the previously solved correct insertion structure (Protein Data Bank [PDB] ID: 4YD1[19], Fig. 2a, b). As observed previously, the Loop1 region is largely disordered in the Complex 1 structure, adopting an alternative conformation compared with that of 1-nt gapped structures (Supplementary Fig. 2a). Despite the nearly identical DNA configuration and the 3′-termini of primer strand to that in the correct insertion structure, a clear deviation was observed at the T:dGMPNPP mispair surrounded by active site residues (Fig. 2b, c). As mentioned above, for the correct insertion opposite

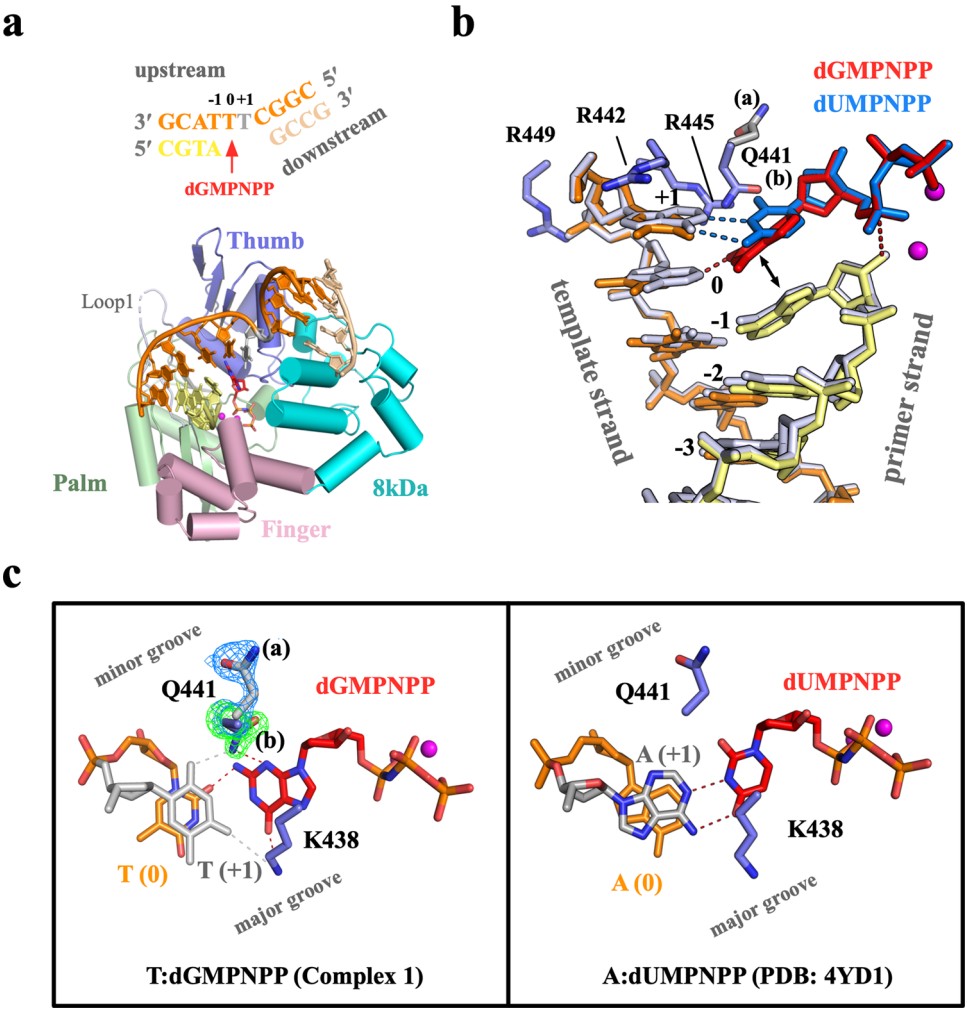

**Fig. 2 Structure of dGTP misincorporation on a 2-nt gapped DNA substrate (Complex 1). a** Overall structure of Pol μ complexed with 2-nt gapped DNA and dGMPNPP. A schematic of the DNA substrate used for crystallization is shown on top with colors corresponding to those in the structures below. Protein domains of Pol μ are labeled and shown in distinct colors. Two catalytic $Mg^{2+}$ and the largely disordered Loop1 region are colored magenta and gray, respectively. The dGMPNPP is shown as a stick (red). **b** Superposition of misincorporation (Complex 1) and correct insertion (PDB: 4YD1) structures of Pol μ complexed with 2-nt gapped DNA. Complex 1 is colored and labeled as in (**a**). The surrounding Arg residues and Gln441 containing two conformations are labeled and shown as sticks. The published structure is colored white with the dUMPNPP in blue. The double arrowhead indicates the stacking between dGMPNPP and 3′-end dA. Hydrogen bonds are indicated by dashed lines. **c** Top views of correct (A:dUMPNPP) and incorrect (T: dGMPNPP) nascent base pairs in (**b**). Lys 438 and Gln441 are shown as sticks. Incoming nucleotides and +1 and 0 templating bases are labeled and colored red, white, and orange, respectively. Two conformations of Q441 are superimposed with the 2Fo–Fc (blue; contoured at 0.8σ) and Fo–Fc omit map (green; contoured at 3.0σ), which were calculated without the conformation B.

a 2-nt gapped DNA, Pol μ skips the first available 3′ templating base (0 positions) but uses the second (+1 position) to maintain the canonical Watson–Crick (WC) pair[19]. However, in Complex 1, dGMPNPP maintains its base stacking with the primer end, making a hydrogen bond with $O_2$ of the first templating T (0 positions) and $N_2$ of dGMPNPP in the minor groove (Fig. 2b, c). The second templating T (+1 position) is stabilized by surrounding Arg residues (Arg442, Arg445, and Arg449) as observed in the correct insertion (Fig. 2b). The T:dGMPNPP mispair fit well in the active site. Lys438 and Gln441 in the major and minor grooves, interact with both dGMPNPP and the second templating T (Fig. 2c). Notably, these two residues form no interactions with the correct paired A:dUMPNPP[19]. These interactions may explain the higher binding affinity of Pol μ incorporating dGTP vs. dATP opposite a 2-nt gapped DNA (Table 1).

**dGTP misincorporation on a 1-nt gapped DNA substrate**. To investigate the mechanism of dGTP misincorporation on 1-nt

gapped DNA, we first solved the binary structure of Pol μ:DNA. This binary complex with template T is almost identical to the published Pol μ:DNA binary complex containing templating A (PDB ID: 4LZG[29]), except for the interactions of the templating base in the active site (Fig. 3a). By replacing dA with dT, the templating T interacts snugly with Lys438 and Gln441 (Fig. 3a), which adopts rotamer conformations similar to those observed in the Complex 1 structure (Fig. 2c). The space between templating T and thumb domain is filled with water and glycerol molecules.

In two WT:DNA ternary structures (Complexes 2 and 3), the template strand displays no distortion during dGTP misincorporation, with the −1 base slightly shifts toward the major groove (Fig. 3b). Despite the weak electron density, dGMPNPP and the 3′-end of the primer strand displays two conformations in both Complex 2 and Complex 3 structures (Fig. 3b, c, Supplementary Fig. 3). In the reaction-ready conformation (conformation A), dGMPNPP is highly distorted and forms a WC geometry with the templating T (Fig. 3b, c). Such distortion is transmitted to the free

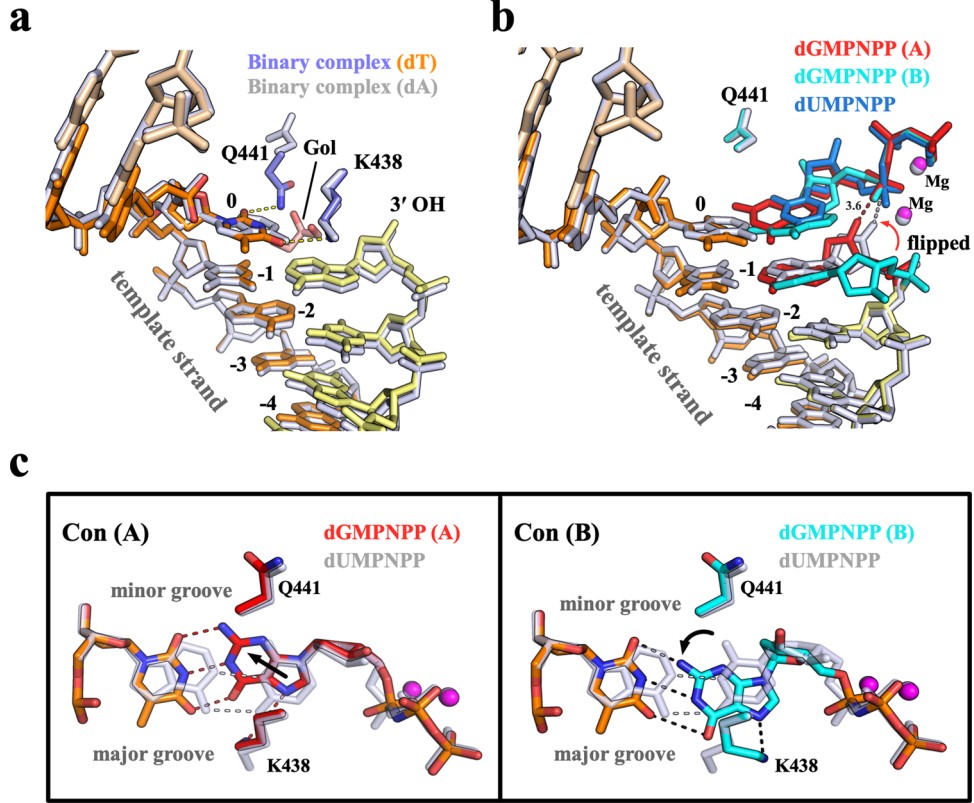

**Fig. 3 Structure of dGTP misincorporation on a 1-nt gapped DNA substrate. a** Superposition of two binary structures of Pol μ complexed with 1-nt gapped DNA containing templating dT (Binary complex) or dA (PDB: 4LZG). The Binary complex solved in the current study has the same coloring and labeling as that in Fig. 2a. The published structure is colored white. **b** Superposition of misincorporation (Complex 2) and correct insertion (PDB: 4M04) structures of Pol μ complexed with 1-nt gapped DNA. The published structure is colored white with dUMPNPP in blue. Complex 2 is colored and labeled as described in (**a**), with two conformations in red (reaction-ready, conformation A) and cyan (flipped, conformation B). The red arrowhead indicates the conformational change required for catalysis. The distance between the 3′-OH and the α-phosphate of the incoming nucleotide is marked by dashed lines. **c** Top views of the two conformations in Complex 3. The nascent base pair (A:dUMPNPP) in the published structure is colored white. Hydrogen bonds are indicated by dashed lines. The black arrowheads indicate the movements of dGMPNPP relative to dUMPNPP.

3′-end dA of the primer strand, which slightly rolls toward the major groove (Fig. 3b). Thus, despite the reasonable distance between reactants (3.6 Å between 3′-OH and α-phosphate), the shifted 3′-OH is not poised for the perfect inline nucleophilic attack during dGTP incorporation, which is consistent with the decreased $k_{cat}$ of dGTP misincorporation compared with the correct insertion (Table 1). In the flipped conformation (conformation B), the sugar and guanine base of dGMPNPP shift toward the major groove, forming a WC-like geometry with templating T (Fig. 3b, c). However, the 3′-end dA of the primer strand is flipped, probably due to stacking with shifted dGMPNPP, with only one weak hydrogen bond being retained between N1 of primer dA and N3 of the templating T (Fig. 3b, Supplementary Fig. 4). As a result, the 3′-OH shifts >10 Å away from the reaction-ready position. In both Complex 2 and Complex 3 structures, the electron density of Lys438 is weak. Gln441 adopts an alternative rotamer conformation toward the finger domain to avoid clashes with the incoming nucleotide (Fig. 3c), which no longer interacts with the T:dGMPNPP mispair (Figs. 2c and 3a).

**Two residues involved in the dGTP misincorporation.** The crystal structures of dGTP misincorporation showed distinct features when template DNA contained a 1 or 2-nt gap, with two closely located residues, Lys438 and Gln441 (Fig. 4a), adopting different rotamer conformations surrounding the T:dGMPNPP

mispair. To further investigate the possible roles of these two residues in dGTP misincorporation, crystal structures of Q441A single-mutant and Q441A/K438A double-mutant proteins complexed with 1-nt gapped DNA and dGMPNPP were determined (Complexes 4 and 5, respectively). In contrast to the stable templating T observed in all the WT:DNA structures (binary complex and Complexes 1-3), the electron density clearly shows an alternative conformation of the templating T and −1 base in the structure of Complex 4 (Fig. 4b, Supplementary Fig. 5). The second conformation of templating T rolls toward the major groove but maintains its interactions with surrounding Arg residues (Fig. 4b). On the other side, the dGMPNPP and 3′-end dA of the primer strand adopt a single conformation, which is very similar to the flipped conformation (conformation B) observed in Complexes 2 and 3 (Fig. 4b, c). The N7 of guanine base of dGMPNPP interacts with Lys438, which forms a distorted WC-like pair with the templating T (Fig. 4c). Although no crystal was obtained for the Pol μ single-mutant K438A, the structure of Q441A/K438A:DNA (Complex 5) revealed shared and distinctive features of dGMPNPP binding compared with the Q441A:DNA structure (Complex 4). While the templating T exhibited a further shift to an alternative conformation, dGMPNPP and the flipped 3′-end dA of the primer strand in Complex 5 adopt a single conformation, similar to that observed in Complex 4 (Supplementary Fig. 6). However, unlike the major-groove-shifting dGMPNPP in Complex4 (Supplementary Fig. 6), dGMPNPP in Complex 5 shifts toward the minor groove and forms a wobble

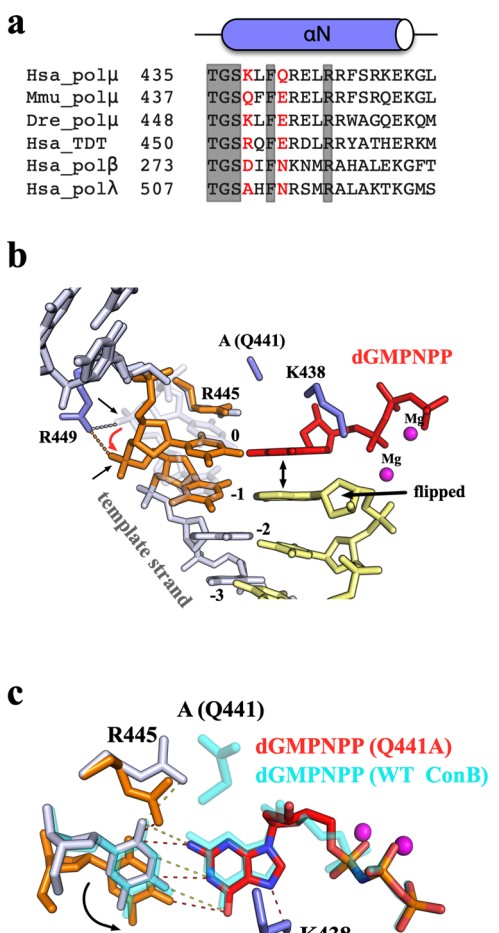

**Fig. 4 Two residues are involved in nucleotide incorporation. a** Sequence alignment of X family polymerases around the αN-helix containing Lys438 and Gln441 (colored red). Human, mouse, and zebrafish proteins are denoted by Hsa, Mmu, and Dre. Identically conserved residues are highlighted in gray. **b** Complex 4 structure. Two conformations of Arg445 and two nucleotides in the template strand (0 and −1 position) are colored white (canonical conformation) and orange (shifted conformation), respectively. The double arrowhead indicates the stacking between dGMPNPP (red) and 3′-end dA. Interactions between Arg449 and the phosphate group of templating T are indicated by dashed lines. Red arrowhead indicates the movement between the phosphate backbone of two DNA conformations. **c** Superposition of Complex 4 (Q441A, colored and labeled as in (**b**)) and the flipped conformation (conformation B, cyan) of Complex 2. Hydrogen bonds are indicated by dashed lines. The black arrowhead indicates the base shifting between the two conformations of the templating T in Complex 4.

base pairing with the second conformation of the templating T (Fig. 5a), which could be explained by the abolished interactions between dGMPNPP and the mutated Lys438 residue (Lys438 in Fig. 4c).

Thus, our structures of Pol μ complexes on a 1-nt gapped DNA (binary complex and Complexes 1–5) reveal important roles of Gln441 and Lys438 in dGTP misincorporation. Kinetic parameters were measured to further test the functional implications from our structures (Table 1). Gln441 from minor groove interacts with templating T in our binary complex and undergoes a rotamer change to avoid clashes with incoming nucleotides regardless of whether correct insertion or misincorporation occurred (Fig. 3b). Compared with WT:DNA structures (Complexes 2 and 3), structures in which alanine was substituted for

Gln441 (Complexes 4 and 5) exhibit a much more stable dGMPNPP (Figs. 4c and 5a), indicating that Gln441 may play a role in discrimination against dGTP misincorporation over the templating T on a 1-nt gapped DNA substrate. This result is consistent with the increased $K_m$ for the correct insertion of dATP but a decreased $K_m$ for dGTP misincorporation of Q441A single-mutant and K438A/Q441A double-mutant proteins (Table 1). In contrast, Lys438 from the major groove interacts with the base pairing of the templating base and incoming nucleotide (T:dGMPNPP in the present study and A:dUMPNPP in published work[29]). Compared with WT, K438A has a much lower catalytic efficiency for both correct (dATP) and incorrect (dGTP) insertions. Moreover, Q441A or K438A single mutant has a higher dGTP misincorporation rate than WT (Table 1), while substitutions of Gln441 (Q441N and Q441E) and Lys438 (K438R and K438Q) retain low dGTP misincorporation rate (Supplementary Table 2). Notably, the Q441A/K438A double mutant displays a remarkably high misincorporation rate of dGTP vs. dATP ($80.4 \times 10^{-2}$).

**In crystallo reaction of dGTP misincorporation.** Steady-state kinetic measurements indicate that the K438A/Q441A double mutant has an overall misincorporation efficiency similar to that of the WT protein (Table 1). To allow the nucleotidyl-transfer reaction, the 3′-OH at the primer end must undergo a large conformational change from the flipped conformation (conformation B) to the reaction-ready conformation (conformation A) (Fig. 3b). Given that only flipped conformation was observed in Complexes 4 and 5 (Fig. 4b, Supplementary Fig. 6), determining whether the reaction truly occurs with such a flipped conformation was of interest. To address this, binary Q441A/K438A:DNA crystals were soaked with Mg$^{2+}$ and dGTP to observe the product of dGTP misincorporation (Post-insertion complex). The electron density at 1.8 Å resolution clearly showed the bond formation between the 3′-OH and incoming dGTP (Fig. 5b). Surprisingly, the nascent dG is pushed out by the second incoming dGTP for the next reaction round, with the newly formed phosphate backbone interacting with a Na$^+$ ion in octahedral coordination (Fig. 5c). Such alkali metal ion binding (Na$^+$ or K$^+$) is frequently observed in not only Pol μ:DNA structures solved by our group and other groups but also published Pol λ and Pol β structures[32,33], which are involved in interactions between upstream DNA and finger domain. In contrast to the minor groove-shifting dGMPNPP observed in its precatalytic ternary structure (Complex 5) (Fig. 5a), dGTP in the Post-insertion complex retained their positions in a manner quite similar to those observed in the Q441A:DNA structure (Complex 4) (Figs. 4c and 5c). This result could be explained by the unexpected interactions between the nascent dG base and the second incoming dGTP in the active site, with N2 of the nascent dG occupying the same position as Lys438 in the Complex 4 structure (Fig. 5c). Nevertheless, the 3′-OH of nascent primer strand was shown to lie outside the active site, which may contribute to its extremely low extension after the T:G mispair (Supplementary Fig. 1).

## Discussion

In addition to homologous recombination, DSBs upon severe DNA damage is largely solved by NHEJ in humans, which is not restricted to a certain phase of the cell cycle[34]. Because of the minimal requirement for aligned DNA ends, NHEJ is considered to be an error-prone pathway. Moreover, NHEJ is essential for the variable–diversity–joining V(D)J recombination process at the early stage of T- and B-cell development[10]. Increasing evidence suggests that X family polymerases containing an

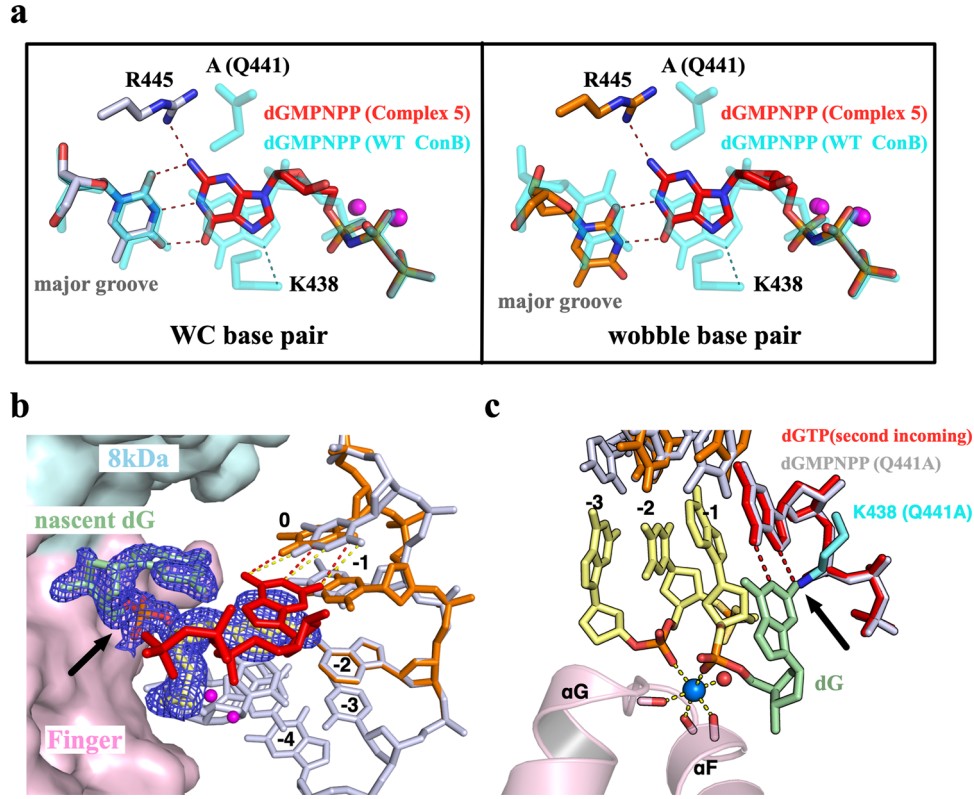

**Fig. 5 In crystallo reaction of dGTP misincorporation. a** Superposition of Complex 5 (colored and labeled as in (Fig. 4b)) and the flipped conformation (conformation B, cyan) of Complex 2. Hydrogen bonds are indicated by dashed lines. **b** Structure of the Post-insertion complex. The protein is presented by the surface as colored in Fig. 2a. The template strand and primer strand are colored orange and white, respectively. The second incoming dGTP is colored red. The nascent dG after the reaction is labeled and colored green, with the newly formed phosphodiester bond indicated by the black arrowhead. Hydrogen bonds are indicated by dashed lines. The electron density of nascent dG is shown in blue with the refined 2Fo–Fc map contoured at 1σ. **c** Side view of the nascent dG in the post-insertion complex (colored and labeled as in (**b**)). The octahedral coordination of K$^+$ ion (blue sphere) is indicated by yellow dashed lines. The water molecule that participates in Na$^+$ ion coordination is shown as a red sphere. For comparison, dGMPNPP and Lys438 in Complex 4 are shown and colored gray and cyan, respectively. Position of NH$_2$ group of nascent dG and Lys438 in the Complex 4 is indicated by the black arrowhead.

N-terminal BRCT domain are involved in the NHEJ[35]. Despite the low-fidelity DNA synthesis incorporating mispaired nucleotides (rNTP or dNTP), Pol μ stands out among human X family polymerases for its distinct activity on noncomplementary DNA ends. Notably, Pol μ can misincorporate dGTP opposite the templating T followed by the high ligation efficiency, which could ultimately lead to genomic instability[22].

In contrast to Pol λ, Pol μ exhibited a much higher catalytic efficiency for the correct insertion opposite 1-nt gapped DNA than that of 2-nt gapped DNA (Table 1). However, Pol μ incorporates dGTP opposite templating T with similar $K_m$ and $k_{cat}$ values, regardless of the 1 or 2-nt gap length of DNA, indicating that no obvious preference exists in the dGTP misincorporation of these two gapped DNA substrates. The crystal structure of Complex 1 containing 2-nt gapped DNA shows that dGMPNPP fits well in the reaction-ready active site, thus making hydrogen bonds with the 3′ templating T (Fig. 2b). Two residues, Lys438 and Gln441, in the major and minor groove interact snugly with dGMPNPP in Complex 1, which is consistent with the retained $K_m$ for dGTP misincorporation but the ~5.5-fold increased $K_m$ for dATP correct insertion on a 2-nt vs. 1-nt gapped DNA substrate (Table 1). Such a configuration substantially differs from the canonical mechanism of insertion of Pol μ and Pol λ on a 2-nt gapped DNA. Pol μ inserts the correct nucleotide opposite the 5′ templating base using the 'skipping ahead' mechanism[19]. In Pol λ structures, the incoming nucleotide pairs with the 3′ templating base with the reoriented 5′ templating base[25]. Given the elevated misincorporation rate of dGTP vs. dATP on 2-nt gapped DNA compared with 1-nt gapped DNA, Pol μ appears to be more mutagenic with unfavored gapped DNA substrates (gap length larger than 1 nt).

The dGTP misincorporation of Pol μ on a 1-nt gapped DNA shows shared and distinctive features compared with that on a 2-nt gapped DNA. The templating T in Complexes 2 and 3 adopts almost the same conformation as the 5′ templating T observed in the Complex 1 structure (+1 position in Fig. 2b). However, because of the limited space of the 1 nt gap, the base of the dGMPNPP swings in the active site, forming a WC-like geometry with templating T in Complexes 2 and 3 (Fig. 3c). As a result, the 3′-end dA of the primer strand shows two conformations, the reaction-ready conformation and the flipped conformation (Fig. 3b). Such a flipped conformation was also observed for the structure of Pol μ incorporation opposite template DNA containing 8OG as reported previously[31]. Despite the reduced fidelity, Mn$^{2+}$ is strongly preferred for dNTP incorporation and mediates a noncanonical reaction for Pol μ, which binds the Mn$^{2+}$/dNTP complex prior to DNA[23]. Comparing two ternary structures in the presence of either Mg$^{2+}$ (Complex 2) or Mn$^{2+}$ (Complex 3) revealed almost identical conformations of dGMPNPP (Supplementary Fig. 2), consistent with their similar $K_m$ values for dGTP misincorporation (Table 1). Thus, the enhanced overall catalytic efficiency of dGTP misincorporation

on a 1-nt gapped DNA could be explained by the intrinsic properties of these two metal ions (e.g., the coordination geometry required for catalysis; the production stabilization role of $Mn^{2+}$ proposed recently[28]). Moreover, given that the guanine base is highly unstable in Complexes 2 and 3, it may explain the preferred pyrimidine misincorporation for 1-nt gapped DNA containing templating T in vitro, which has a smaller size compared with guanine (Fig. 1).

Gln441 interacts with the unpaired 5' templating T in both binary complex and Complex 1 structures and undergoes a rotamer change toward the finger domain to avoid clashes with base-pairing after incoming nucleotide binding, which is observed in all the normal and misincorporation ternary structures solved by our group and others[19,23,28–30]. Kaminski et al.[31] recently demonstrated a similar conformation of Gln441 during adenine insertion opposite template DNA containing 8OG. Based on the interactions with templating 8OG, Gln441 is proposed to be involved in the incoming nucleotide base selectivity. This is consistent with our structural observations and kinetic measurements. The templating T is stable as its normal position in the presence of Gln441 in Complexes 2 and 3, which causes the destabilized dGMPNPP and the 3'-end dA of the primer strand. In contrast, dGMPNPP is stable when Gln441 is replaced with Ala, leading to the destabilized templating strand (Complexes 4 and 5). Given that Q411A vs. WT exhibited both increased dGTP binding affinity and the elevated relative misincorporation rate on a 1-nt gapped DNA (Table 1), Gln441 may function as the fidelity checkpoint in the minor groove by stabilizing the templating base. However, such interrogation should meet the criteria for a normal 1-nt gapped DNA substrate, which in turn would promote unexpected incorporation (e.g., dGTP opposite 2-nt gapped DNA or adenine opposite the templating 8OG). Lys438 on the other side adopts various rotamer conformations while interacting with the base pair in the major groove (Fig. 3c). Despite the swinging of the dGMPNPP base in the active site, T:dGMPNPP retains a WC-like geometry in WT:DNA structures (Complexes 2 and 3). Similar WC-like mispairs were also found in the structures of other X family polymerases (e.g., Pol λ[33]), in contrast to the wobble conformation present in Y family polymerases[36]. Given that wobble T:dGMPNPP was observed in the structure of the double mutant (Complex 5), these two residues in the major and minor grooves appear to interact with the templating base and fine-tune the active site microenvironments of the incoming nucleotide binding.

The T:G mispair in WC-like geometry is considered to be a potential promutagenic event causing genome instability[37,38]. Unlike Pol λ and Pol β, the T:G mispair after Pol μ's dGTP misincorporation could be effectively ligated by NHEJ ligases[22]. To investigate the potential mechanism, we attempted to determine the structures of WT and mutant Pol μ containing the T:G mispair in the active site, either by crystallization of the Pol μ-product DNA binary complex or soaking of the Pol μ-substrate DNA binary complex in a solution containing $Mg^{2+}$/dGTP. However, the T:G mispair appeared to be extremely unstable in the crystals, and the poor electron density did not allow for model building. Nevertheless, a post-insertion structure was solved by soaking the K438A/Q441A-DNA complex with $Mg^{2+}$/dGTP, and this structure differed from all the Pol μ product structures previously reported. Unexpectedly, after the first round of reaction, a second dGTP was shown to occupy the nucleotide-binding pocket and form WC-like base pairing with the templating T (Fig. 5b, c). Moreover, this dGTP was stabilized by hydrogen bond interactions with nascent G extrahelically interacting with the finger domain, which mimics the interactions between dGMPNPP and Lys438 in the major groove (Fig. 5c).

## Methods

**Protein expression and purification**. Polymerase core domain (residues 136–494) with Loop2 deletion (Δ398–410) of WT and mutant human Pol μ was synthesized and cloned into pET28a plasmid. *Escherichia coli* BL21(DE3) harbors this plasmid were grown at 37 °C in LB medium containing 40 μg ml⁻¹ Kanamycin to an $OD_{600}$ of 0.6–0.8. Protein expression was induced at 30 °C for 5 h by the addition of 0.4 mM isopropyl-β-D-thioga-lactopyranoside (IPTG). cells were harvested and resuspended in buffer A (25 mM Tris/HCl, pH 8.0, 500 mM NaCl, 5% glycerol, 3 mM β-ME, protease inhibitors), lysed by sonication. After centrifugation, the supernatant was purified by nickel affinity HP column (GE Healthcare) equilibrated with buffer A, washed with the same buffer containing 45 mM imidazole, and finally eluted with 300 mM imidazole. After desalting, the protein was purified on HiTrap Heparin HP column (GE Healthcare) using linear NaCl gradient from 0.15 M to 1 M in 25 mM Tris/HCl, pH 8.0, 0.1 mM EDTA, 5% glycerol. The protein was finally purified on Superdex75 10/30 column (GE Healthcare) with buffer C (25 mM Tris/HCl, pH 7.5, 100 mM KCl, 1 mM DTT) and stored at −80 °C.

**Crystallization of Pol μ with gapped DNA**. Template DNA, upstream primer and 5'-phosphorylated downstream primer at a 1:1.2:1.2 molar ratio were annealed in 100 mM Tris/HCl, pH 7.5 by heating for 5 min at 98 °C and slow cooling to 4 °C (Supplementary Table 3). The freshly purified protein (~7 mg/ml) and DNA were mixed at a 1:1.5 molar for crystallization. Crystallization drops contained 1 μl of a good solution and an equal volume of Pol μ-DNA complex. All the crystals were grown by sitting-drop vapor-diffusion method over wells containing various conditions: Binary complex crystals were grown in 18% (v/v) 2-propanol, 0.1 M sodium citrate (pH 5.5) and 16–20% (w/v) polyethylene glycol (PEG) 4000; Complex 1 crystals were grown in 0.2 M potassium sodium tartrate tetrahydrate, 0.1 M Bis–Tris/HCl (pH 6.5) and 10–15% (w/v) PEG10000; Complexes 2 and 3 crystals were grown in 0.1 M Tris/HCl (pH 8.0) and 16–20% PEG3350; Complex 4 and post-insertion crystals were grown in 2% (v/v) 1,4-Dioxane, Tris/HCl (pH 8.0) and 16-21% PEG3350; and Complex 5 crystals were grown in 5% (v/v) hexylene glycol, 0.1 M HEPES (pH 7.5) and 10–15% (w/v) PEG 10000. Ternary complex crystals were obtained by soaking the binary crystals with 2 mM dGMPNPP (or 10 mM dGTP for Post-insertion complex) and 10 mM $MgCl_2$ (or 5 mM $MnCl_2$ for Complex 3) at 4 °C for 4 h. The crystals were flash-frozen in liquid nitrogen after stepwise soaking in the reservoir solution containing 10, 20, and 30% glycerol. Diffraction data were collected on beamline BL17U at Shanghai Synchrotron Radiation Facility (Shanghai, China) and were integrated and scaled with the XDS suit[39]. The structures were determined by molecular replacement using a published pol μ structure (PDB ID: 4M04) as the search model. Structures were refined using PHENIX[40] and interspersed with manual model building using COOT[41]. The statistics for data collection and refinement are listed in Supplementary Table 1. All residues are in the most favorable and allowed regions of the Ramachandran plot. All structural figures were rendered in PyMol (www.pymol.org).

**Kinetic measurements**. The primer extension assay and steady-state $K_m$ and $k_{cat}$ measurements were carried out by using a 6-FAM-5'-labeled upstream primer, 5'-phosphorylated downstream primer, and template strand containing 1 or 2 nt gaps. The DNA sequences and nucleotides used in these assays are shown in Supplementary Table 2. For the reaction condition, usually Pol μ (5–200 nM) was incubated with 6-FAM-5'-labeled gapped DNA substrates (250 nM) and various concentrations of dATP or dGTP (0–300 μM for dATP and 0–500 μM for dGTP) in 50 mM Tris/HCl (pH 7.5), 1 mM DTT, 0.1 mg/mL bovine serum albumin, 5% glycerol, and 10 mM $MgCl_2$ or $MnCl_2$. Reactions were carried out at 37 °C for 5, 7, or 9 min and quenched by 10 times formamide loading buffer (80% deionized formamide, 10 mM EDTA, pH 8.0, 1 mg/ml xylene cyanol), heated to 98 °C for 10 min, placed on ice, and products resolved on 20% polyacrylamide sequencing gels containing 7 M urea. Quantification and curve fitting was done as described before[36]. All reactions were independently repeated at least three times. Uncropped and unprocessed scans of Fig. 1 and Supplementary Fig. 1 are supplied in the Source Data file.

**Reporting summary**. Further information on research design is available in the Nature Research Reporting Summary linked to this article.

## Data availability
The coordinates and structure factors have been deposited to Protein Data Bank with accession codes 7CO6 (binary complex), 7CO8 (Complex 1), 7CO9 (Complex 2), 7COA (Complex 3), 7COB (Complex 4), 7COC (Complex 5), and 7COD (post-insertion complex). Other data are available upon request. PDB files used for structure comparisons (Figs. 2 and 3: 4YD1, 4LZG, and 4M04) are from the Protein Data Bank (https://www.rcsb.org). Source data are provided with this paper.

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

## Acknowledgements

We would like to thank the staff at the Shanghai Synchrotron Radiation Facility (SSRF in China) for their assistance in data collection. This work was supported by the National Key Research and Development Program of China (2017YFA0503900); the National Natural Science Foundation of China (31670819, 31500656, 31670065, and 31870051); and Zhejiang Provincial Natural Science Foundation for Outstanding Young Scientists (LR16C050002).

## Author contributions

Y.Z. and Y.H. conceived the project. M.G., Z.C., J.H., and J.D. carried out the protein purification. M.G., Yu W., and Y.T. carried out crystallization and data collection. Y.Z. and M.G. determined crystal structures and analyze the data. M.G., Yi W., and Y.T. did biochemical experiments and kinetics analyses. L.W., B.T., and H.X. gave technical support and conceptual advice. Y.Z. and M.G. wrote the paper. All authors discussed the results and commented on the paper.

## Competing interests

The authors declare no competing interests.
