## [Peer Review File · Nature Communications]

REVIEWER COMMENTS

Reviewer #1 (Remarks to the Author):

The authors selected an important problem for investigation, and the results described are timely in this field. Pol μ is capable of performing gap-filling repair synthesis in the nonhomologous end joining pathway. The misincorporation of dGTP opposite templating T by Pol μ is well known, as is the production of the pro-mutagenic T:G mispair. In the present manuscript, crystal structures of human Pol μ inserting dGTP on 1 and 2 nt gapped DNA substrates containing templating Ts are described. Two active site amino acids, K438 and Q441, were chosen for detailed study including their substitution with alanine. The authors concluded that these two amino acids in the wild-type enzyme make important functional interactions with the nascent T:dGTP base pair and play distinct roles in promoting dGTP misincorporation. The structures themselves are well described in this manuscript, and they are of high resolution, enabling detailed interpretations; the authors are commended for these features.

Critique:

1) The overall interpretations that 438 and 441 play roles in dGTP insertion, along with atomic level explanations, far exceed the level of information at hand. The interpretations in this regard should be extensively modified in a revised version of the manuscript. In most cases a simple descriptive statement about the structures would enhance this manuscript and make it more professional.

2) The kinetic analysis reported failed to demonstrate strong effects of the 438 and 441 side chains. This undercuts the overall conclusions of the present manuscript, essentially as noted above. Therefore, comments about any functional roles of both of these side chains should be revised, so to make the comments far more guarded. An additional concern about the kinetic results is that the assays do not appear to have been under true steady-state conditions: the 25:1 ratio of substrate to enzyme is, of course, appropriate, but the time course of product formation was not provided. This is a concern because in the gel images shown substrate consumption appears to have been large enough to compromise the linearity of the product production vs time relationship. Can the authors provide evidence that product production was linear as a function of time for the data shown in Table 1?

Reviewer #2 (Remarks to the Author):

Previously a dGTP:dT mismatch and a 8-oxo-dGTP:dT mismatch insertion coupled with ligation to characterize the promutagenic DNA repair intermediate were reported by Sam Wilson's group (Nat. Comm. 2018, ref. 22). The structural basis for a skip-ahead property for which Pol μ skipped the first available templating base and formed a correct base pair with the 2nd templating base near the 5'P end of the down stream primer was reported by Moon et al. in 2015 PNAS, ref 19). However, the structural basis for the dGTP:dT mismatch catalyzed by Pol μ on a 1-nt and 2-nt gapped DNA has been lacking. In this manuscript entitled "Mechanism of genome instability mediated by human DNA polymerase mu misincorporation", Guo et al. provided the structural basis for the dGTP:dT mismatch catalyzed by Pol μ on a 1-nt and 2-nt gapped DNA dGTP:dT mismatch by solving seven crystal structures involving Pol μ : Binary complex of Pol μ with a 1-nt gap DNA with templating dT, five ternary complexes each containing a dGMPNPP:dT mismatch, and a post-insertion complex of Q441A/K438A. The authors also performed steady-state kinetic measurements to report the base selection and efficiency of Pol μ dNTP incorporation opposite a templating dT on a 1-nt and 1-nt gapped DNA with Mg²⁺ or Mn²⁺. The novelty of this work is the structural basis for the dGMPNPP:dT mismatch on a 1-nt and a 2-nt gapped DNA. With the structures solved, the authors then identified two critical dGTP:dT-interacting residues, Lys438 and Gln441, and went further to determine the dGMPNPP:dT mismatches- containing mutant structures as well steady-state kinetic data for the wild type and the mutant enzymes (Table 1). Interestingly, for a 2-nt gapped DNA, different from a correct base pair reported by Moon et al. (PNAS 2015) that skipped the first available templating base, the structures reported in this work showed that Pol μ incorporated dGMPNPP to the first templating base to form a dGMPNPP:dT mismatch. Overall, the combined structural basis for the dGTP:dT mismatch on a 1-nt and a 2-nt gapped DNA and the kinetic data reported in this manuscript advances the understanding of the

structure-function relationship of Pol μ in terms of NHEJ reaction and the possible genome instability due to the misincorporation. Therefore, this work is recommended to be published in Nature Communications after minor revision.

Minor comments:

- (1). Line 281: cite ref. 22 again here
- (2). Line 311: add ref. 23 again here
- (3). A small table summarizing the all of the structural numbers, in particular complexes 1- 5, in the main text will be helpful, although this information is available in Table S1.
- (4). Kcat should be changed to kcat; Kpol should be changed to kpol. Small k instead of capital K is usually used to describe rates.

Reviewer #1:

The authors selected an important problem for investigation, and the results described are timely in this field. Pol μ is capable of performing gap-filling repair synthesis in the nonhomologous end joining pathway. The misincorporation of dGTP opposite templating T by Pol μ is well known, as is the production of the pro-mutagenic T:G mispair. In the present manuscript, crystal structures of human Pol μ inserting dGTP on 1 and 2 nt gapped DNA substrates containing templating Ts are described. Two active site amino acids, K438 and Q441, were chosen for detailed study including their substitution with alanine. The authors concluded that these two amino acids in the wild-type enzyme make important functional interactions with the nascent T:dGTP base pair and play distinct roles in promoting dGTP misincorporation. The structures themselves are well described in this manuscript, and they are of high resolution, enabling detailed interpretations; the authors are commended for these features.

Author's response: We are grateful to the reviewer for appreciating the work and suggestions for improving the work quality. In the revised version, additional kinetic measurements for these two residues were provided to strengthen the conclusion. And we have carefully revised the text to avoid over-interpretation of the functions of these two residues.

Critique:

I) The overall interpretations that 438 and 441 play roles in dGTP insertion, along with atomic level explanations, far exceed the level of information at hand. The interpretations in this regard should be extensively modified in a revised version of the manuscript. In most cases a simple descriptive statement about the structures would enhance this manuscript and make it more professional.

Author's response: Thank you for the comment and suggestion. In the revised version, we further explored the functions of these two residues in the dGTP misincorporation by measuring the kinetic parameters of four additional mutant proteins (K438R, K438Q, Q441N, and Q441E). Please find our response to "critique 2" below. Through the revised text, we carefully revised our statements to avoid over-interpretations, e.g., "Two residues involved in the dGTP misincorporation (Page 10 Line 193)" instead of "Two residues play distinct roles in nucleotide discrimination". The last paragraph in discussion section was deleted. Sentences concerning proposed functions of these two residues (especially about the side chains) were removed/revised. Words such as "distinct residues", "unique residues", and "discriminating" were rephrased. Supplementary tables were also updated in the revised text.

2) The kinetic analysis reported failed to demonstrate strong effects of the 438 and 441 side chains. This undercuts the overall conclusions of the present manuscript, essentially as note above. Therefore, comments about any functional roles of both of these side chains should be revised, so to make the comments far more guarded. An additional concern about the kinetic results is that the assays do not appear to have been under true steady-state conditions: the 25:1 ratio of substrate to enzyme is, of course, appropriate, but the time course of product formation was not provided. This is a concern because in the gel images shown substrate consumption appears to have been large enough to compromise the linearity of the product production vs time relationship. Can the authors provide evidence that product production was linear as a function of time for the data shown in Table I?

Author's response: We agree with your point that more experimental data were needed to demonstrate the effects of the Lys438 and Gln441 side chains. Based on the sequence alignments of pol μ proteins as well as other X family polymerases (Figure 4a), four additional kinetic parameters of pol μ mutants (K438R, K438Q, Q441N, and Q441E) were measured (please see updated supplementary TableS2 below). In our structures, Lys438 from major groove adopts various rotamer conformations while interacting with the T:dGMPNPP. Consistent with that of WT protein, both K438R (with similar side chain property) and K438Q (Q440 in mouse pol μ) retain low misincorporation rate of dGTP vs. dATP (2.5% and 1.9%, respectively) compared with that of the K438A protein (15.4%). Similar results were also observed for Q441N (N279 and N513 in human Pol β and Pol λ , respectively) and Q441E (in other pol μ proteins) mutant proteins, which had low misincorporation rate of dGTP vs. dATP compared with that of the Q441A protein. Hence these results strengthen the conclusion that Lys438 and Gln441 are involved in the dGTP misincorporation of pol μ . Nevertheless, more experiments (e.g., structure determination if possible) are required to elucidate the exact function of the side chains of these two residues, as you mentioned. And we have revised text concerning the functional roles of these side chains by descriptive statement. It has been mentioned in the revised text as following: "Moreover, Q441A or K438A single mutant has a higher dGTP misincorporation rate than WT (Table 1), while substitutions of Gln441 (Q441N and Q441E) or Lys438 (K438R and K438Q) retains low dGTP misincorporation rate (Table S2)."

Enzyme	dNTP	k_m (μM)	k_{cat} (min^{-1})	k_{pol} (k_{cat}/k_m) ($\text{min}^{-1}\cdot\mu\text{M}^{-1}$)	Relative k_{pol}	Misincorporation rate
WT	dATP	3.1 \pm 0.3	236.2 $\times 10^{-3}$ \pm 5.3 $\times 10^{-3}$	76.6 $\times 10^{-3}$	1.00	6.4%
	dGTP	5.7 \pm 0.5	28.3 $\times 10^{-3}$ \pm 0.6 $\times 10^{-3}$	4.9 $\times 10^{-3}$	0.06	
Q441E	dATP	8.6 \pm 0.6	398.5 $\times 10^{-3}$ \pm 8.6 $\times 10^{-3}$	46.3 $\times 10^{-3}$	0.60	2.2%
	dGTP	79.2 \pm 9.5	79.8 $\times 10^{-3}$ \pm 2.8 $\times 10^{-3}$	1.0 $\times 10^{-3}$	0.01	
Q441N	dATP	3.6 \pm 0.5	188.6 $\times 10^{-3}$ \pm 7.1 $\times 10^{-3}$	52.4 $\times 10^{-3}$	0.68	5.2%
	dGTP	4.5 \pm 0.7	12.0 $\times 10^{-3}$ \pm 0.4 $\times 10^{-3}$	2.7 $\times 10^{-3}$	0.04	
K438R	dATP	2.8 \pm 0.3	354.0 $\times 10^{-3}$ \pm 7.6 $\times 10^{-3}$	126.4 $\times 10^{-3}$	1.65	2.5%
	dGTP	12.0 \pm 1.8	37.9 $\times 10^{-3}$ \pm 1.0 $\times 10^{-3}$	3.2 $\times 10^{-3}$	0.04	
K438Q	dATP	2.0 \pm 0.2	93.2 $\times 10^{-3}$ \pm 1.6 $\times 10^{-3}$	46.6 $\times 10^{-3}$	0.61	1.9%
	dGTP	1.9 \pm 0.1	1.7 $\times 10^{-3}$ \pm 0.02 $\times 10^{-3}$	0.9 $\times 10^{-3}$	0.01	

The gel image of supplementary Figure 1 showed the extension after the TG mispair by pol μ . However, because of the very weak mispair-extension activity as mentioned in the main text, an increased DNA:protein ratio (25:1) was used for the reactions compared with those for the Figure 1 (50:1). For kinetic measurements, we kept the production ratio below 20% for all the reactions, which ensured the linearity of the product production vs. time relationship (similar to our previous studies on human Pol η , ref 6 and 36). Our results for correct dATP insertions are comparable to published values obtained with DNA containing varied sequence contexts (Moon et al. in 2015 *PNAS*, ref 19). A typical gel for kinetic measurements of Table 1 is shown below.

Thank you again for reviewing our manuscript and your valuable suggestions to improve our manuscript.

Reviewer #2 (Remarks to the Author):

Previously a dGTP:dT mismatch and a 8-oxo-dGTP:dT mismatch insertion coupled with ligation to characterize the promutagenic DNA repair intermediate were reported by Sam Wilson's group (*Nat. Comm.* 2018, ref. 22). The structural basis for a skip-ahead property for which Pol μ skipped the first available templating base and formed a correct base pair with the 2nd templating base near the 5' P end of the downstream primer was reported by Moon et al. in 2015 *PNAS*, ref 19). However, the structural basis for the dGTP:dT mismatch catalyzed by Pol μ on a 1-nt and 2-nt gapped DNA has been lacking. In this manuscript entitled "Mechanism of genome instability mediated by human DNA polymerase mu misincorporation", Guo et al. provided the structural basis for the dGTP:dT mismatch catalyzed by Pol μ on a 1-nt and 2-nt gapped DNA dGTP:dT mismatch by solving seven crystal structures involving Pol μ : Binary complex of Pol μ with a 1-nt gap DNA with templating dT, five ternary complexes each containing a dGMPNPP:dT mismatch, and a post-insertion complex of Q441A/K438A. The authors also performed steady-state kinetic measurements to report the base selection and efficiency of Pol μ dNTP incorporation opposite a templating dT on a 1-nt and 1-nt gapped DNA with Mg^{2+} or Mn^{2+} . The novelty of this work is the structural basis for the dGMPNPP:dT mismatch on a 1-nt and a 2-nt gapped DNA. With the structures solved, the authors then identified two critical dGTP:dT-interacting residues, Lys438 and Gln441, and went further to determine the dGMPNPP:dT mismatches-containing mutant structures as well steady-state kinetic data for the wild type and the mutant enzymes (Table 1). Interestingly, for a 2-nt gapped DNA, different from a correct base pair reported by Moon et al. (*PNAS* 2015) that skipped the first available templating base, the structures reported in this work showed that Pol μ incorporated dGMPNPP to the first templating base to form a dGMPNPP:dT mismatch. Overall, the combined structural basis for the dGTP:dT mismatch on a 1-nt and a 2-nt gapped DNA and the kinetic data reported in this manuscript advances the understanding of the structure-function relationship of Pol μ in terms of NHEJ reaction and the possible genome instability due to the misincorporation. Therefore, this work is recommended to be published in *Nature Communications* after minor revision.

Author's response: We are grateful to the reviewer for appreciating the work and suggestions for improving the work quality.

Minor comments:

(1). Line 281: cite ref. 22 again here

Author's response: Thank you. It has been cited.

(2). Line 311: add ref. 23 again here

Author's response: Thank you. It has been cited.

(3). A small table summarizing the all of the structural numbers, in particular complexes 1- 5, in the main text will be helpful, although this information is available in Table S1.

Author's response: Thank you for the suggestion. In the revised version, additional table summarizing all the structures (revised Table 2) was provided.

Structure name	Protein	Gapped DNA	Metal	dNTP	PDB Code
Binary complex	WT	1-nt gap	Mg ²⁺	-	7CO6
Complex 1	WT	2-nt gap	Mg ²⁺	dGMPNPP	7CO8
Complex 2	WT	1-nt gap	Mg ²⁺	dGMPNPP	7CO9
Complex 3	WT	1-nt gap	Mn ²⁺	dGMPNPP	7COA
Complex 4	Q441A	1-nt gap	Mg ²⁺	dGMPNPP	7COB
Complex 5	K438A/Q441A	1-nt gap	Mg ²⁺	dGMPNPP	7COC
Post-insertion complex	K438A/Q441A	1-nt gap	Mg ²⁺	dGTP	7COD

(4). Kcat should be changed to kcat; Kpol should be changed to kpol. Small k instead of capital K is usually used to describe rates.

Author's response: Corrected, thank you.

REVIEWERS' COMMENTS

Reviewer #1 (Remarks to the Author):

Oxidized nucleotides are formed during numerous biological processes such as in bacterial and viral infection, inflammation, cancer, and upon exposure to oxidatively damaging conditions in the cellular environment. Oxidized nucleotides cause mutagenesis and are inserted into genomic DNA during replication and repair eliciting polymerase discrimination to prevent cancer and disease. How oxidized nucleotides are inserted during double strand break repair is unknown. This is significant as mutagenic double strand break repair is associated with biological processes such as development of antibodies against varied antigens, neuronal differentiation and development, telomere maintenance, and gene editing.

In this work, the authors reveal high-resolution time-lapse X-ray crystallography snapshots of double strand break repair polymerase μ undergoing DNA synthesis with undamaged and oxidized nucleotides, providing mechanistic insight into the structural basis for mutagenic repair in these processes.

Overall, the results improve the quality of work in this field. Regarding the manuscript there are some inconsistent statements, English editing is required and removal of excessive use of definite article "the."

CONCERNS

-Differences in kinetics expressed as a % don't say much other than G is inserted poorly opposite T and the fold differences cited are not significant (e.g., 3.5-fold); the kinetic tables with "relative k_{pol} " are strange; seems like some of the data in Fig S1 are not consistent with other results such as the increased "efficiency" of K438R for dATP insertion opposite template T; Fig1/S1, there is a band beyond the first nucleotide in the lane where insertions were performed with dGTP, but this just means the enzyme fills the 2 nt gap and the same band is present in all the other lanes, so what is the relevance of this? The kinetic workup is not strong enough to consider differences; experiments employed 10 mM, not 1 mM Mn, which has been shown to inhibit; please explain.

-Fig 2b the coloring of the two conformations with different colors and also consistently within Fig2 and other figures; lines 159-161 delete; line 164 ".binary structure of Pol μ :DNA" add "with template T". These items should be corrected.

-More simulated annealing omit Fo-Fc density should be shown to support the varied conformations

-Did they make the change of K438 to Ala or Asp as per pol beta/lambda and characterize these variants to substantiate the claim regarding the role of K438? I suggest asking the authors to change statements about the role of 438, because the evidence is not yet strong enough. I don't see how one can conclude that K438 is responsible for "mutagenic incorporation" based on the overall result that indicate a combination of factors is at play.

Reviewer #2 (Remarks to the Author):

The comments and suggested minor revisions of this reviewer have been addressed satisfactorily. However, I found a new minor issue but not sure if this is a new one or one I missed in the previous review. In Table 1, the catalytic efficiency is defined as k_{pol} , which could be confusing since k_{pol} is commonly used in polymerase pre-steady state kinetics to represent the equivalent of k_{cat} . In steady-state kinetics, the catalytic efficiency is commonly just represented by k_{cat}/K_m .

Reviewer #1:

Oxidized nucleotides are formed during numerous biological processes such as in bacterial and viral infection, inflammation, cancer, and upon exposure to oxidatively damaging conditions in the cellular environment. Oxidized nucleotides cause mutagenesis and are inserted into genomic DNA during replication and repair eliciting polymerase discrimination to prevent cancer and disease.

How oxidized nucleotides are inserted during double strand break repair is unknown. This is significant as mutagenic double strand break repair is associated with biological processes such as development of antibodies against varied antigens, neuronal differentiation and development, telomere maintenance, and gene editing.

In this work, the authors reveal high-resolution time-lapse X-ray crystallography snapshots of double strand break repair polymerase μ undergoing DNA synthesis with undamaged and oxidized nucleotides, providing mechanistic insight into the structural basis for mutagenic repair in these processes.

Overall, the results improve the quality of work in this field. Regarding the manuscript there are some inconsistent statements, English editing is required and removal of excessive use of definite article “the.”

Author’s response: Thank you for all the comments and suggestions. We have carefully revised the text including removal of excessive use of definite article “the”.

CONCERNS

-Differences in kinetics expressed as a % don’t say much other than G is inserted poorly opposite T and the fold differences cited are not significant (e.g., 3.5-fold); the kinetic tables with “relative k_{pol} ” are strange; seems like some of the data in Fig S1 are not consistent with other results such as the increased “efficiency” of K438R for dATP insertion opposite template T;

Author’s response: Thank you for the comments and suggestions. We agree with your point that % and “relative k_{pol} ” might not be a good way to describe the relative efficiency of misincorporation by Pol μ . In the revised text and kinetic tables, all the “ K_{pol} ” and “relative k_{pol} ” columns were removed. And “misincorporation rate” was replaced by “ f_{mis} ”, expressing as the ratio of $[k_{cat}/K_{mdGTP}]/[k_{cat}/K_{mdATP}]$.

Fig1/S1, there is a band beyond the first nucleotide in the lane where insertions were performed with dGTP, but this just means the enzyme fills the 2 nt gap and the same band is present in all the other lanes, so what is the relevance of this?

Author’s response: Yes, in both Fig1 and FigS1, a weak band beyond the product of dGTP misincorporation (for both 1 nt and 2 nt gapped DNA substrates; indicated by red arrowheads in FigS1) was observed with increased protein concentration. However, such band was barely observed after dATP, dCTP or dTTP incorporation,

indicating that Pol μ has weak activity to extend the primer after a T:G mispair *in vitro*.

The kinetic workup is not strong enough to consider differences; experiments employed 10 mM, not 1 mM Mn, which has been shown to inhibit; please explain.

Author's response: Thank you for the comment. To our knowledge, the inhibitory of Mn catalyzed dNTP incorporation by Pol μ was experimentally observed by Luis Blanco group (ref 27 in revised text), which has different experimental design compared to the current work. For example, different protein/DNA concentrations and DNA substrate (gapped DNA instead of NHEJ substrate was used in the current work) were used for our kinetic measurements. And the inhibitory effect is actually depending on the types of NHEJ substrate. Moreover, 10 mM Mn was present in the soaking solutions to obtain Pol μ -DNA-dNTP crystals by our group and other groups (e.g., ref 28), which is consistent with Mn concentration for our kinetic measurements.

-Fig 2b the coloring of the two conformations with different colors and also consistently within Fig2 and other figures; lines 159-161 delete; line 164 “..binary structure of Pol μ :DNA” add “with template T”. These items should be corrected.

-More simulated annealing omit Fo-Fc density should be shown to support the varied conformations

Author's response: The color of alternative conformation in Fig2b and the mentioned items have been revised. Moreover, the Fo-Fc density of Gln441 with alternative conformation has been added in the revised figure to support the varied conformation (please see revised Fig2 below).

-Did they make the change of K438 to Ala or Asp as per pol beta/lambda and characterize these variants to substantiate the claim regarding the role of K438? I suggest asking the authors to change statements about the role of 438, because the evidence is not yet strong enough. I don't see how one can conclude that K438 is responsible for "mutagenic incorporation" based on the overall result that indicate a combination of factors is at play.

Author's response: Thank you again for the comment. We have already shown that compared with WT, K438A has a much lower catalytic efficiency for both correct (dATP) and incorrect (dGTP) insertions but elevated relative efficiency of dGTP misincorporation (Table 1). However, we were unable to obtain K438D protein (no expression) so far. We agree with your point that whether Lys438 is directly involved in the "mutagenic incorporation" remains a question. Nevertheless, we believe that this residue should be involved in the interaction with the base pair (between templating base and incoming nucleotide) during dNTP incorporation. In the revised text, we have carefully rephrased sentences concerning the function of Lys438 to avoid over-interpretation.

Thank you again for reviewing our manuscript and your valuable suggestions to improve our manuscript.

Reviewer #2 (Remarks to the Author):

The comments and suggested minor revisions of this reviewer have been addressed satisfactorily. However, I found a new minor issue but not sure if this is a new one or one I missed in the previous review. In Table 1, the catalytic efficiency is defined as k_{pol} , which could be confusing since k_{pol} is commonly used in polymerase pre-steady state kinetics to represent the equivalent of k_{cat} . In steady-state kinetics, the catalytic efficiency is commonly just represented by k_{cat}/K_m .

Author's response: Thank you again for good suggestion. In the revised text and tables, the catalytic efficiency was represented by k_{cat}/K_m .